# NAD(P)H Quinone Oxidoreductase-1 Expression Promotes Self-Renewal and Therapeutic Resistance in Non-Small Cell Lung Cancer

**DOI:** 10.3390/genes14030607

**Published:** 2023-02-28

**Authors:** Brian Madajewski, Michael A. Boatman, Ivan Martinez, Julia H. Carter, Erik A. Bey

**Affiliations:** 1Memorial Sloan Kettering-Cornell Center for Translation of Cancer Nanomedicine, Memorial Sloan Kettering Cancer Center, 415 East 68th Street, New York, NY 10065, USA; 2West Virginia University Cancer Institute, West Virginia University, 1 Medical Center Drive, Morgantown, WV 26506, USA; 3Wood Hudson Cancer Research Laboratory Inc., 931 Isabella Street, Newport, KY 41701, USA

**Keywords:** NQO1, self-renewal, tumor-initiating cells, chemo-resistance, non-small cell lung cancer, proliferation, spheroids

## Abstract

Identifying cellular drivers responsible for enhancing cancer cell resistance to therapeutics provides critical information for designing more effective drugs. Populations of slowly growing, self-renewing, chemo-resistant cells purportedly contribute to the development of therapeutic resistance in many solid tumors. In the current study, we implemented a tumor spheroid model to determine whether NAD(P)H quinone oxidoreductase-1 (NQO1) was requisite for self-renewal and promotion of the drug-resistant phenotype in non-small cell lung cancer (NSCLC). We found that stable depletion of NQO1 in A549 and H358 human NSCLC tumor models inhibits self-renewal capabilities, as demonstrated by a reduced ability to form primary, secondary, and tertiary spheroids. In contrast, the rescue of NQO1 expression restored the tumor cells’ ability to form spheroids. Additionally, we discovered that NQO1 depletion renders cisplatin-refractory tumor spheroids highly susceptible to drug treatment. Together, these results suggest that NQO1 loss reduces the self-renewing capabilities of NSCLC cells and enhances their susceptibility to clinically relevant therapeutics. These findings describe a novel role for NQO1 and suggest that combining NQO1-inhibitors with conventional chemotherapeutics may enhance anti-tumor effects.

## 1. Introduction

Lung cancer has long been the leading cause of cancer-related deaths in the U.S. and worldwide [1]. In 2023, it is estimated that more than 127,000 deaths will occur in the U.S. due to cancer of the lung and bronchus [2]. Small cell lung cancer (SCLC) will comprise 20% of estimated new lung cancer cases; 80% will be non-small cell lung cancer (NSCLC) [3,4,5]. Both NSCLC and SCLC have abysmal 5-year survival rates of 15% and 5%, respectively [1,5]. The significant factors contributing to the overall poor survival rates observed in lung cancer patients include chemo-resistance, late-stage diagnosis, and the subsequent metastatic spread of disease [6,7]. 

Interestingly, overexpression of cytoprotective genes reduced the therapeutic effects of commonly prescribed chemotherapy drugs and radiation in several NSCLC cases [8]. NAD(P)H: quinone oxidoreductase 1 (NQO1) is overexpressed in ~60% of all NSCLC cases [9,10]. NQO1 is a phase II detoxifying enzyme that neutralizes dangerous intracellular quinones and scavenges reactive oxygen species (ROS) [11,12]. NQO1 overexpression has recently been investigated as a therapeutic target in many malignancies. Treatment with the quinone analog, β-lapachone (ARQ-761), has shown promise in inducing tumor-specific cell death. ARQ-761 initiates a futile cycle with NQO1, increasing intracellular ROS, hyperactivating poly ADP ribose polymerase, and depleting nucleotide depletion, eventually leading to cell death [13,14,15]. While substantial data exists to support targeting NQO1 overexpression, the mechanism by which NQO1 promotes the tumorigenic phenotype is not well understood.

In addition to NQO1 overexpression, other contributing factors drive tumor development; for example, cancer stem cells (CSCs), also known as tumor-initiating cells (TICs), initially described by Bonnet and Dick (1997) [16,17]. They identified a small leukemic cell population, defined by a set of cluster differentiation markers (CD34^+^CD38^−^), able to establish disease at minimal numbers compared with the bulk tumor population [18], and CSCs have since been characterized in other malignancies, including breast, brain, and prostate cancers [19,20,21]. This cell population initiates tumor growth, circumvents conventional therapies, and metastasizes to distant locations; thus, targeting it is a promising strategy for improving patient survival [22]. 

To better understand the role of NQO1 in tumorigenesis and tumor maintenance, we investigated the relationship between NQO1 and the CSC phenotype (Figure 1). We previously found that tumor NQO1 expression levels promoted many malignant characteristics, including tumorigenesis, anoikis resistance, and cellular invasion [23]. NQO1 also demonstrated a novel role as a regulator of activity for the purported CSC marker, aldehyde dehydrogenase (ALDH) [23]. To expand upon these novel findings, we present new evidence linking NQO1 to the promotion of self-renewal and chemotherapeutic resistance in models of NSCLS. NQO1 is necessary for in vitro serial spheroid formation, resistance to cisplatin treatment, and in vitro limiting-dilution tumor formation. 

Our data clearly demonstrate, for the first time, that NQO1 is vital to maintenance of the CSC population, and that interventions focused on reducing NQO1 expression or activity levels in solid tumors, in combination with standard therapeutics, may provide better control of disease and improved patient outcomes.

## 2. Materials and Methods

### 2.1. Cell Culture

A549, H358, and H596 cell lines were grown in Dulbecco’s Modified Eagles Media (DMEM) supplemented with 10% fetal bovine serum (FBS) and 1% L-glutamine. Cell lines were cultured at 37 °C with 20% oxygen and 5% carbon dioxide before spheroid culture. Cells were passaged weekly and supplemented with fresh media.

### 2.2. Stable NQO1 Protein Knockdown

A549 and H358 cell lines underwent stable NQO1 knockdown, as previously described [23]. The shNQO1 retroviral vector (Open Biosystems, RHS1764-9691437), and lentiviral particles (Santa Cruz Biotechnology, sc-37139) were used to generate stable knockdown cell lines (shNQO1, shNQO1 (lenti)) and empty vector controls (shCtr, shCtr (lenti)) were created for both A549 and H358 cell lines by viral infection in polybrene supplemented media containing either shNQO1 or shCtr viral particles. Following viral infection, cells were placed in limited dilution under puromycin (2 µg/mL) selection and screened for NQO1 expression via Western blot. A549-shNQO1 and H596 cell lines were forced to express NQO1 via a retroviral vector (LPC-NQO1), or were transduced with an empty vector control (LPC-X) as previously described [23]. Transduced cells were then placed in limited dilution for selection and evaluated for NQO1 expression via Western blot.

### 2.3. Western Blots

Protein lysates were separated by 10% SDS-PAGE and transferred onto a PVDF membrane. Membranes were blocked with 5% milk in PBST (0.1% Tween-20 in PBS) for 1 h at room temperature, and then incubated overnight with β-actin (1:5000, Santa Cruz Biotechnology (Dallas, TX, USA), to determine loading equivalence of each sample) at 4 °C. The blots were then washed in PBST and incubated for 1 h with 1:5000 dilution of goat anti-mouse IgG-HRP in 5% milk in PBST. This process was then repeated using various antibodies at different dilutions, including a 1:5000 dilution of monoclonal NQO1 antibody (clone A-180, Santa Cruz Biotechnology), as well as 1:1000 dilutions of monoclonal Shh, SOX2, and Nanog antibodies (Cell Signaling, Danvers, MA, USA). Pierce ECL Western blotting substrate (Thermo Scientific, Waltham, MA, USA) was used to visualize bands on Hyblot-CL autoradiography film (Denville Scientific, South Plainfield, NJ, USA).

### 2.4. NQO1 Activity Assay

NQO1 enzyme activity was performed as previously described [23]. Briefly, 2 × 10^7^ cells of each cell line were collected, and pellets were solubilized in extraction buffer for 20 min, after which they were centrifuged at 18,000× *g* for 20 min at 4 °C. Supernatants were collected into Eppendorf tubes and stored at −80 °C. Samples were run according to the manufacturer’s protocol for the NQO1 activity assay kit (Abcam). Results were read at an absorbance of 440 nm every 20 s for 5 min, utilizing the Synergy-H1 Hybrid microplate reader. Plates were shaken both before and after each reading.

### 2.5. Spheroid Formation

Cell culture plates (Corning, Corning, NY, USA) were coated twice with 0.2% poly-hema/95% ethanol, incubated at 60 °C overnight, and allowed to dry. Plates were washed twice with milli-Q water immediately before use. Cells were trypsinized and treated with trypsin neutralizing solution (1:1 ratio) before being counted using a hemocytometer. One hundred sixty thousand cells were plated in 0.25% FBS DMEM supplemented with 1% L-glutamine. Cells were allowed to form spheroids over 14 days, at which time they were collected, trypsinized into single-cell suspensions, and used for respective assays. For spheroids grown in methylcellulose suspension, 1% methylcellulose in 0.25% FBS DMEM was further diluted at a ratio of 1:1 in 0.25% FBS DMEM, and cells were added to this mixture. The cell suspension was plated on low attachment plates for 2 weeks, followed by image acquisition and quantification.

### 2.6. Extreme Limited Dilution Assay

Low attachment 96-well plates were prepared by treating plates with 0.2% poly-HEMA in 95% ethanol and allowing them to dry overnight. This process was repeated a second time to ensure uniform coating. Before plating, 96-well plates were washed twice with sterile milli-Q water. Cells were trypsinized, counted, and plated in 0.25% FBS-containing DMEM at densities of 40, 120, 360, and 720 cells per well. Each dilution was performed in a total of 24 wells. Cells were incubated at 37 °C, 20% O_2_, 5% CO_2_ and allowed to expand over three weeks (21 days), at which time the wells were examined for the presence of spheroids. A well containing a spheroid was counted as a positive well; multiple spheroids per well did not increase the number of positive wells. The number of positive wells per dilution was then entered into the extreme limited dilution cancer stem cell frequency calculating software available http://bioinf.wehi.edu.au/software/elda/ (accessed on 15 March 2015).

### 2.7. Drug Treatment Studies

Spheroids were collected after 14 days and trypsinized into single-cell suspension. Cells were counted and suspended in DMEM media supplemented with 0.25% FBS and 1% L-glutamine at a concentration of 10,000 cells/mL. A total of 200 µL of cell suspension was added to each well. Cells were allowed to attach overnight, and the following day were treated with 0, 2.5, 5, and 10 µM cisplatin dissolved in DMSO. Stock concentrations of 2.5, 5, and 10 mM cisplatin were diluted 1:1000 in 0.25% FBS DMEM supplemented with 1% L-glutamine and incubated overnight. Each dose was performed in 8 replicates for each dose tested. Twenty-four hours after initial treatment, cell viability was assessed using Cell-Titer Glo (Promega, Madison, WI, USA), according to the manufacturer’s protocol utilizing the Synergy-H1 Hybrid Reader.

### 2.8. Cell Proliferation Assays

Spheroids were mechanically (pipetting) and enzymatically (trypsin) broken down into single-cell suspension. Cell suspension was then treated 1:1 with trypsin neutralizing solution to inactivate the trypsin. Cells were quantified using a hemocytometer and suspended in 0.25% FBS DMEM at a concentration of 10,000 cells/mL. In 96-well plates, 100 µL of each cell suspension was plated in 5 wells, and the respective plate was collected at the time points of 0, 24, 48, and 72 h. At this time, 1 × 10^6^ cells were also collected in a micro-centrifuge tube to generate a standard curve. Collected plates were washed once with PBS, aspirated, and froze at −80 °C until all time points were collected. Plates were quantified using a CyQuant Cell Proliferation Assay (Thermo Fisher, Waltham, MA, USA), according to the manufacturer’s protocol utilizing the Synergy-H1 Hybrid Reader.

### 2.9. Quantitative Real-Time qRT-PCR

Total RNA was isolated utilizing the Trizol extraction method [24]. cDNA was created from the total RNA sample, utilizing an iScript cDNA Synthesis Kit (Bio-Rad, Hercules, CA, USA). qRT-PCR was performed using primers designed for NQO1 (Fwd-5′-CCAGATATTGTGGCTGAACAAA-3′; Rev-5′-TCTCCTATGAACACTCGCTCAA-3′), and SsoAdvanced Universal SYBR Green Supermix (Bio-Rad). Samples were analyzed using the CFX Connect Real-time qRT-PCR System (Bio-Rad). Relative expression values were calculated utilizing double delta C_t_ analysis.

### 2.10. Statistical Analysis

Data analyses were performed using GraphPad Prism 6 software. Statistical significance was determined by using Student *t*-tests, and *p* values from these analyses were reported. Differences were considered significant when *p* values were <0.05.

## 3. Results

### 3.1. NQO1 Is Essential for In Vitro Spheroid Formation

One of the hallmarks of the transformed phenotype is cellular proliferation in the absence of attachment [25,26,27,28]. To determine the role of NQO1 in perpetuating CSC growth, we used spheroid formation assays as an assessment tool [29]. Stable retroviral expression of NQO1 shRNA reduced NQO1 levels in A549 and H358 NSCLC cell lines (Figure 2A,C, respectively). As expected, the reduction in NQO1 protein expression correlated with a decrease in NQO1 activity (Figure 2B,D). This reduction in NQO1 protein levels led to a near-total loss of primary spheroid formation in A549 (Figure 3A) and H358 (Figure 3C) cell lines. A549-shCtr cells generated ~30 spheroids per field of view (50X magnification), whereas the A549-shNQO1 population demonstrated a significantly reduced spheroid number (Figure 3B). Interestingly, the inability of A549-shNQO1 cells to form spheroids was not a result of significant cell death, as trypan blue exclusion assays and analysis of apoptotic endpoints (PARP-1 cleavage and AIF expression) showed no significant difference over time (Appendix A). These data were recapitulated using a second, lentiviral-driven shRNA toward NQO1 (Appendix A).

The sphere-forming capability of the H358 cell line was reduced in comparison with A549 cells. This may be due to the overall lower level of NQO1 activity in H358 cells in comparison with A549 cells (Figure 2D,B, respectively). Despite the low number of spheroids formed by H358 cells, however, a significant difference exists between the H358-shCtr and H358-shNQO1 populations (Figure 3D).

To confirm spheroid formation resulted from clonal expansion, as opposed to aggregation, we performed spheroid formation studies using media containing 0.5%, and found a similar significant difference in spheroid-forming ability in A549 cells with depleted NQO1 protein levels versus controls (Appendix A).

Lastly, to confirm the importance of NQO1 activity in the formation of spheroids, we treated A549-shCtr and H358-shCtr cell cultures with the NQO1 inhibitor dicoumarol (50 μM). Inhibiting NQO1 had the same effect as knocking down its expression, with significant reductions in the number of spheroids formed by A549 and H358 cells in the presence of the inhibitor (Appendix A).

### 3.2. Primary Spheroids Increase Expression of Stem Cell-Related Genes and NQO1

Next, we validated the role of cancer stem cell markers in spheroid formation by evaluating the expression of the purported stem cell markers, Sox2, Shh, and Nanog [30,31]. Culturing A549 cells as spheroids increased the expression of each marker in comparison with attached-culture conditions. This increased expression of Sox2, Shh, and Nanog could be reversed by plating dissociated A549 spheroids in differentiation-inducing media containing 10% FBS (Appendix A). These results illustrate that spheroids generated by NQO1-expressing cells also express known stem cell markers, suggesting that NQO1 contributes to the maintenance of the CSC-like population.

Interestingly, in primary spheroids, NQO1 protein levels in the A549-shNQO1 population remained reduced in comparison to A549-shCtr spheroid cells (Appendix A), as expected; however, NQO1 mRNA expression, as assessed by qRT-PCR, was highly upregulated in the A549-shNQO1 spheroid population in comparison with cells grown in traditional 2D culture (A549-shNQO1 2D; Appendix A). This suggests that knocking down NQO1 drives cells to compensate by increasing NQO1 gene expression in spheroid culture. The small proportion of A549-shNQO1 cells that can form spheroids show a robust increase in NQO1 mRNA expression in comparison to identical cells grown under attached conditions. Therefore, NQO1 is likely a significant contributor to cell viability and proliferative capacity under anchorage-independent conditions.

### 3.3. NQO1 Is Necessary for Serial In Vitro Sphere Formation

CSCs divide asymmetrically to continually replenish themselves while also producing a population of proliferative progenitors [32]. These can promote tumor initiation, even after gross reduction of the overall tumor cell population. Serial tumor sphere formation assays, from primary to tertiary sphere formation, were performed to identify a tumor-initiating cell population within the A549 and H358 cell lines. In support of our earlier data (Figure 3), secondary and tertiary spheroid formation was also markedly reduced in A549-shNQO1and H359-shNQO1 cells (Figure 4A and Figure 4B, respectively).

Interestingly, the impact of NQO1 loss on serial spheroid formation was more severe in H358 cells than in A549 cells, which naturally express higher NQO1 at a higher level. This result indicates that a threshold level of NQO1 expression and activity may be necessary to support self-replication and growth in anchorage-independent settings. In support of this, a second H358-shNQO1 clone (H358-4C20), which demonstrated a higher level of NQO1 expression than H358-shNQO1, maintained the ability to form primary spheroids equal to the level observed in controls (Appendix A).

Together, these data strongly suggest a role for NQO1 in anchorage-independence and, thus, tumor cell growth and proliferation. Additionally, significant decreases in the formation of serial spheroids in both the A549 and H358 NSCLC cell lines suggest that NQO1 plays a supportive role in perpetuating the population of tumor-initiating cells.

### 3.4. NQO1 Expression Rescues Spheroid Formation and Enhances Tumor-Initiating Cell Frequency

To confirm our findings, we performed NQO1 rescue experiments, re-expressing NQO1 in A549-shNQO1 via retroviral vector at a similar level to its expression in parental A549 cells (Figure 4C). A significant increase in the number of spheroids formed was observed in comparison with A549-shNQO1 cells transduced with the control vector (Figure 4D). Furthermore, we re-expressed NQO1 in the H596 NSCLC cell line, which contains the *2 NQO1-inactivating polymorphism [14]. Spheroids formation per field of view in these cells was also increased in comparison with empty vector controls (H596 LPC-X; Figure 5). These results again indicate that NQO1 is necessary to support sphere-forming capabilities across multiple NSCLC cell lines.

Limited dilution assays demonstrate tumor-initiating cell frequency within a cell population [28]. To determine their frequency in the A549 and H358 cell population, we subjected each line to an extreme limiting dilution assay. A549 cells expressing NQO1 have an approximately 12-fold increase in tumor-initiating cells in comparison with A549 cells lacking NQO1 expression (A549-shNQO1). This finding also held for H358 cells, to a lesser extent (~ 2-fold increase) (Table 1). Overall, these results support the suggestion that NQO1 is vital for NSCLC cells to grow in anchorage-independent conditions and supports the maintenance of the tumor-initiating cell population [23].

### 3.5. NQO1 Knockdown Inhibits Cell Proliferation and Increases Chemotherapeutic Resistance of NSCLC Tumor Spheroids

Cancer, by definition, is a disease of uncontrolled cellular proliferation [33]. To better understand the role of NQO1 in promoting cancer cell proliferation, we dissociated spheroid cultures and monitored their growth in two-dimensional culture. The expression of NQO1 significantly increased the proliferative capacity of both A549 and H358 cells when compared to their respective NQO1 knockdown clones over time (Figure 6A and Figure 6B, respectively).

One hallmark of the CSC phenotype is their inherent resistance to chemotherapeutic challenges [34]. Given that NQO1 has also been shown to protect malignant cells against chemotherapy [8], as well as to function as a protective agent against chemotherapy-induced toxicities in normal cells [35], we hypothesized that spheroids expressing NQO1 would be resistant to cisplatin-induced cell death [8,36,37,38,39]. To test this, we dissociated spheroids into single-cell suspensions and challenged them with increasing concentrations (0, 2.5, 5.0, or 10 µM) of cisplatin for 24 h. A dose-dependent decrease in cellular viability occurred in the cell lines with transduced with shNQO1 vectors versus non-targeted control cell lines (Figure 6C,D). These data suggest that the expression of NQO1, and its corresponding activity, are important factors driving the resistance of NSCLC cells to chemotherapeutics. These findings also indicate that NSCLC may be made susceptible to cisplatin, and other chemotherapeutics, by implementing combinatorial NQO1 depletion or inhibition strategies.

## 4. Discussion

It is well-known that cellular populations within a tumor are heterogeneous and serve various roles to promote the survival of the bulk tumor [40,41]. One purportedly critical population within the overall tumor milieu is the tumor-initiating cell, or cancer stem cell. This group of cells is thought to be responsible for giving rise to chemo-resistance, tumor recurrence, metastasis, and ultimately reduced patient prognosis [42,43,44]. This specialized population of cells is understood to have features similar to those of normal stem cells, including the ability to differentiate into diverse cell types. Thus, tumor stem-like cells enable their continued self-renewal while also replenishing other necessary tumor populations. Many scholarly endeavors have been made over the past thirty years to identify and characterize this sought-after cell type across numerous tumor types in hopes of developing new therapeutic strategies for cancer therapy [45,46,47].

The identification and isolation of the CSC population from bulk tumor populations has primarily been accomplished using cellular markers to distinguish a fraction of cells capable of recreating the heterogenous bulk tumor [48]. Although no iron-clad definition of the stem-like population across tumor types has been developed, many studies have developed protocols using these various stem cell markers to identify, isolate, and test this unique cell population. For example, in breast cancer, the most common markers used in the isolation of the stem-like cells include CD44^high^CD24^low^ populations, as well as populations that demonstrate high levels of ALDH^high^ activity and SOX2 expression [49,50]. In lung cancer, ALDH^high^ activity, along with Notch expression, is closely linked to this population in patients whose tumors tend to become refractory to therapy, recur, and metastasize [51,52].

To enrich the stem-like populations, researchers have developed various tissue culture and propagation methodologies. For example, the 3-dimensional tumor spheroid model has become a staple amongst assays used to enrich for the stem-like cells in culture, as well as a methodology for evaluating drug efficacy [16]. Spheroid models are believed to be more representative of tumor conditions encountered in vivo, making them ideal tools to test the efficacy of therapeutic compounds [53,54].

We have shown that NQO1 depletion in the general population of A549 and H292 NSCLC tumor cells is correlated with the loss of ALDH^high^ activity [23]. We also demonstrated that NQO1 depletion inhibited proliferation, invasion, and growth in vivo. Those data were the first to indicate that the level of NQO1 expression in malignancies was linked to tumor initiation and growth. Results demonstrating a loss of ALDH^high^ activity following reduction of NQO1 led us to postulate that NQO1 may be necessary for maintaining the CSC population.

Here, we compared the tumor spheroid-forming ability of NSCLC cells with or without the expression of NQO1. It is noteworthy that the cell lines chosen in this manuscript, including A549 (an NQO1 positive cell line), H358 (an NQO1 positive cell line), and H596 (an NQO1 negative cell line), all have wild type EGFR expression. EGFR expression is thought to regulate several essential CSC properties, including metabolism and dormancy [55]. The NQO1 negative cell line, H596, has wild type RAS expression, while A549 and H358 have mutated RAS. The p53 status is different in all three cell lines, where A549 is wild type, H358 is null, and H596 has a p53 mutation. In both shNQO1 knockdown models (A549 and H358), we found that NQO1 depletion reduced the ability of NSCLC cells to form primary tumor spheroids. We also observed severe impairment of the ability of shNQO1 cells to form secondary and tertiary spheroids, likely a result of a diminished TIC, or CSC population. Interestingly, when NQO1 expression was rescued in these knockdown cell lines, tumor spheroid formation was restored. In addition to rescue experiments, we also examined the NQO1 null cell line, H596, by driving expression of NQO1 and again testing the cells’ ability to generate spheroids. Results from this study demonstrated a significant increase in the spheroid-forming ability of H596 cells expressing NQO1 versus the NQO1-null parental line.

In support of NQO1 enhancing the stem-like cell population, we demonstrated that spheroid culture enhances the expression of known stem cell markers SOX2, Shh, and Nanog. Interestingly, these results were observed in both our A549-shCtr and A549-shNQO1 cell lines. When evaluating the expression of NQO1 in cells cultured as spheroids, we noticed an increased, yet still significantly reduced, expression of NQO1 in A549-shNQO1 cells in comparison to control. This suggests that, for A549-shNQO1 to survive and expand under spheroid conditions, increased expression of NQO1 is vital. We postulate that the increased expression of NQO1 in our knockdown line is the reason we observed limited spheroid-forming ability in A549-shNQO1 cells and an increase in the aforementioned stem cell markers during spheroid culture. These data, coupled with the loss of ALDH^high^ activity in our previous work, strongly suggest that NQO1 plays a supportive role in the maintenance of cancer stem-like cell population in NSCLC.

Lastly, we investigated the effect that NQO1 depletion had on chemotherapeutic resistance using tumor spheroids. Our data demonstrated that shCtr spheroids were resistant to cisplatin, while spheroids from cells transduced with the shNQO1 vector were more susceptible to treatment across tested concentrations. These data imply that the depletion of NQO1 expression increases the susceptibility of NSCLC cells to chemotherapeutic intervention.

In summary, our data provide a sound rationale for developing therapeutics for tumors that overexpress NQO1. Designing drugs that will either (1) inhibit the expression of NQO1 or (2) reduce its overall activity could prove promising. Loss of NQO1 and its activity will limit a cancer cell’s ability to survive in detached conditions, thus impacting a tumor cell’s metastatic potential. Reducing NQO1 levels will also help to drive down the CSC population found in tumors. Lastly, NQO1-directed therapeutics will increase a tumor cell’s susceptibility to clinical chemotherapeutics. Combining novel NQO1-targted therapeutics with currently available, effective treatments may result in synergistic combinations that have yet to be appreciated. These combinatorial approaches hold promise to manage disease better and improve overall patient outcomes.

## Figures and Tables

**Figure 1 genes-14-00607-f001:**
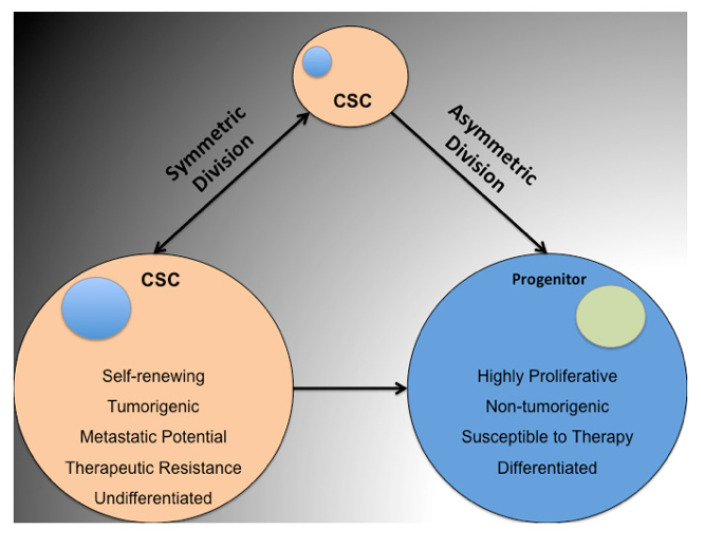
Diagram depicting the lineage of cells produced by cancer stem cells and the phenotypic differences between the cell types they produce.

**Figure 2 genes-14-00607-f002:**
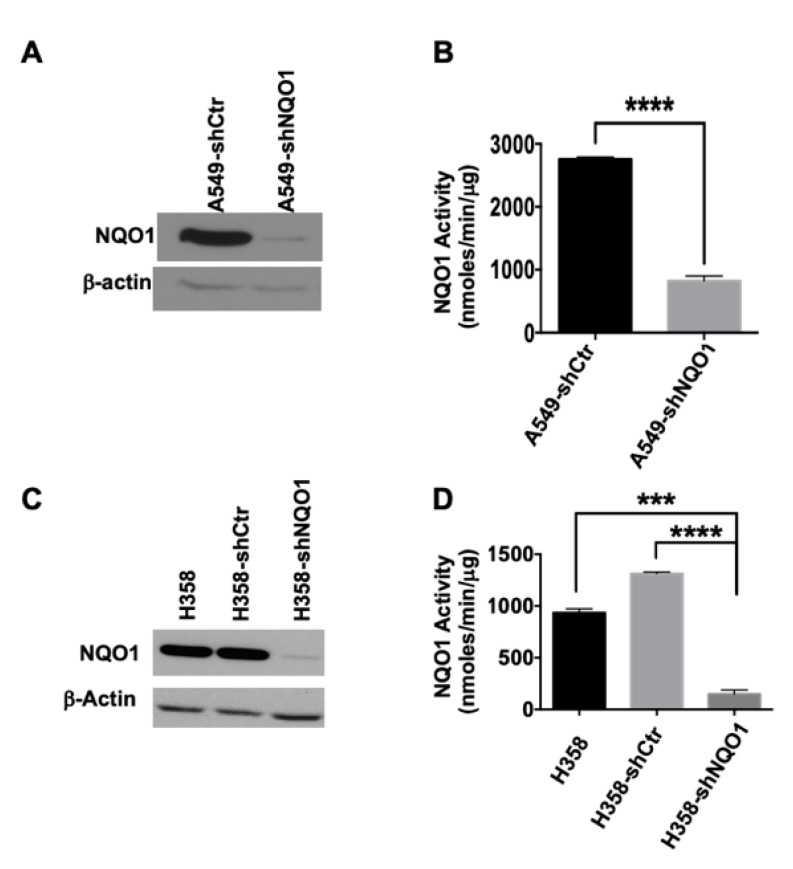
Stable knockdown of NQO1 expression in NSCLC cells reduces NQO1 activity. In (**A**), A549 lung adenocarcinoma cells were evaluated for NQO1 expression following retroviral transduction of shRNA targeted at NQO1 (A549-shNQO1), or empty vector control (A549-shCtr). In (**B**), A549-shCtr and A549-shNQO1 cell lines were analyzed for NQO1 activity, where loss of NQO1 protein expression correlated with a significant decrease in NQO1 activity (**** *p* < 0.0001). In (**C**), the lung cancer cell line, H358 was evaluated for NQO1 protein expression following retroviral transduction of shRNA directed toward NQO1 (H358-shNQO1), or the empty vector control (H358-shCtr). In (**D**), H358-shCtr and H358-shNQO1 cell lines were evaluated for NQO1 activity and demonstrated that loss of NQO1 protein expression correlated with a significant decrease in NQO1 activity (**** *p* < 0.0001; *** *p* < 0.001).

**Figure 3 genes-14-00607-f003:**
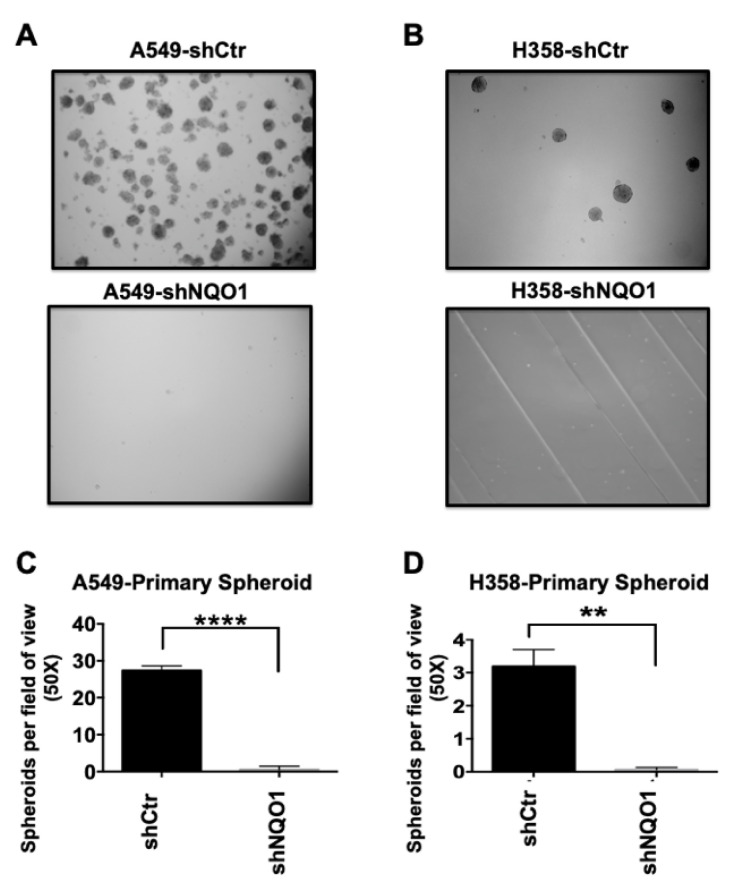
NQO1 depletion reduces primary tumor spheroid formation. (**A**,**B**) are representative images of primary spheroid formation in both A549 and H358 cell lines, respectively. A549-shCtr cells (**A**, **top panel**) robustly form primary spheroids in comparison to the A549-shNQO1 cell lines (**A**, **bottom panel**). In the H358 cell lines, again the H358-shCtr cell line (**B**, **upper panel**) shows an increase in the number of primary spheroids formed in comparison to H358-shCtr (**B**, **lower panel**). Primary spheroid counts are quantified for A549 and H358 in (**C**) and (**D**), respectively (**** *p* < 0.0001; ** *p* = 0.0035).

**Figure 4 genes-14-00607-f004:**
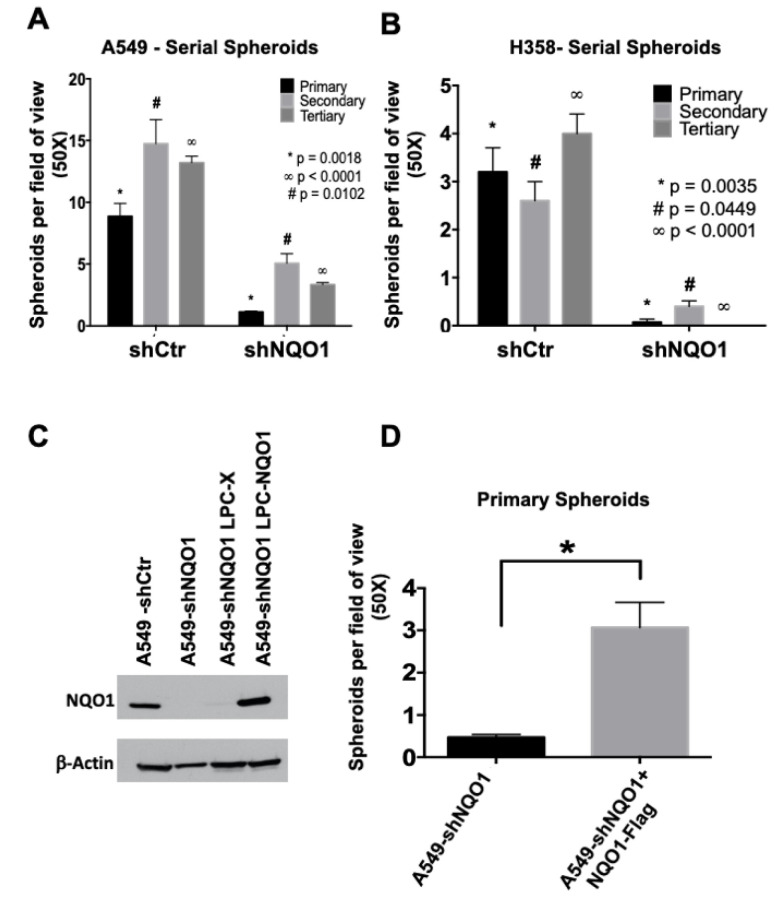
Loss of NQO1 expression reduces serial spheroid formation, and re-expression of NQO1 rescues tumor spheroid-forming ability. The self-renewal capabilities of lung cancer stem cells were assayed via a serial spheroid formation assay. In (**A**), A549-shCtr cells form similar numbers of spheroids in primary (black bars), secondary (light grey bars), and tertiary generations (dark grey bars), where the A549-shNQO1 cells form significantly fewer spheroids in each generation (* *p* = 0.0018, # *p* = 0.0102, ∞ *p* < 0.0001). The H358 cell line, as seen in (**B**), shows similar results (* *p* = 0.0035, # *p* = 0.0449, ∞ *p* < 0.0001). (**C**) shows the Western blot analysis of A549-shNQO1 cells transduced with the retroviral NQO1 vector in an effort to induce re-expression of NQO1 protein. (**D**) shows primary spheroid formation in NQO1 re-expressing A549-shNQO1 cells. The rescued NQO1 expression induced a significant increase in the formation of primary spheroids (* *p* = 0.0121).

**Figure 5 genes-14-00607-f005:**
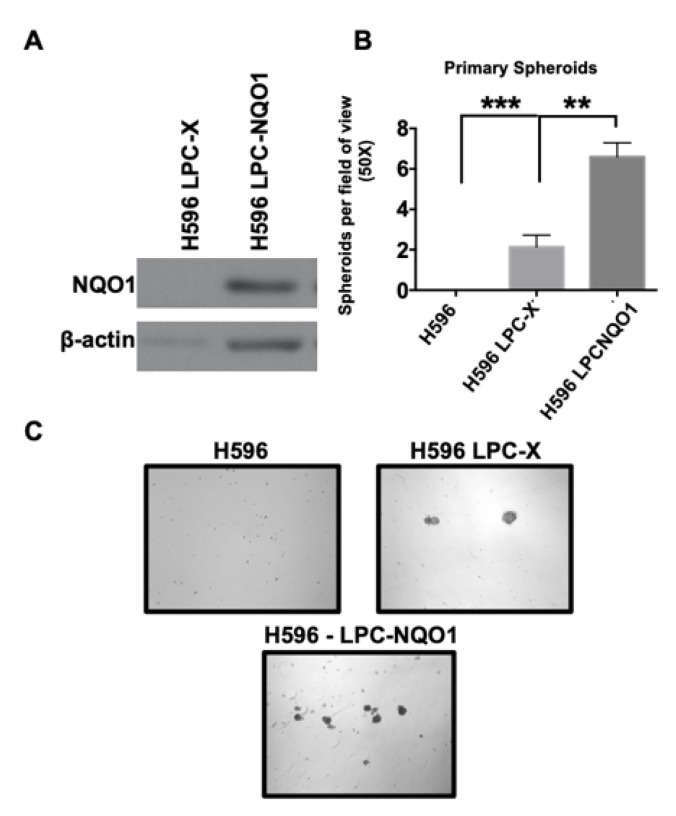
Overexpression of NQO1 in NQO1 null H596 cells enhances tumor spheroid formation. The NQO1-null lung cancer cell line, H596, was forced to over-express NQO1 via retroviral vector and subjected to the spheroid formation assay. NQO1 expression in the LPC-NQO1 cell line was confirmed via Western blot in (**A**). In (**B**), the numbers of spheroids formed per field of view were quantified in the parental, LPC-X, and LPC-NQO1 H596 cell lines. (**C**) shows representative images corresponding to the quantification found in (**B**) (*** *p* = 0.0007; ** *p* = 0.0078).

**Figure 6 genes-14-00607-f006:**
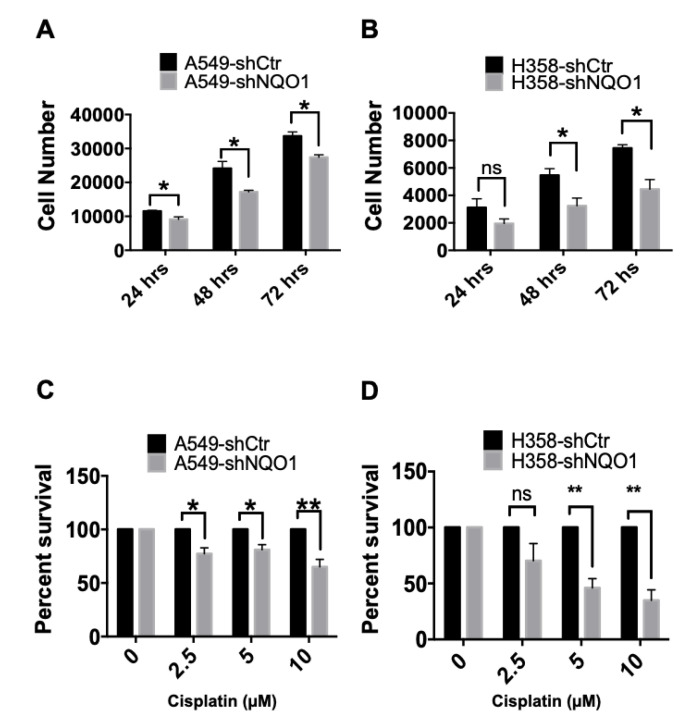
NQO1 depletion increases sensitization to cisplatin treatment and inhibits spheroid-cultured cell proliferation. In (**A**), A549-shCtr and A549-shNQO1 primary spheroids were dissociated, plated, and subsequently treated with cisplatin the following day at the given concentrations. Cell viability was computed by comparing the percentage of A549-shNQO1 survival to that of the A549-shCtr cell line. After 24 h of cisplatin exposure, there was a significant difference in the surviving fraction of cells in the A549-shCtr population in comparison to the A549-shNQO1 cell line (* *p* < 0.05; ** *p* = 0.0076). In (**B**), an identical experiment was carried out comparing the H358-shCtr and H358-shNQO1 cell lines, with similar results. H358-shNQO1 cells were more susceptible to cisplatin treatment than the H358-shCtr cell line (** *p* < 0.005). In (**C**), A549-shCtr and A549-shNQO1 cells grown in spheroid conditions were assayed for cellular proliferation. Spheroid cells were plated, collected at the specified time points, and enumerated using the CyQuant Cell Proliferation Assay (Thermo Fisher). At each time point, a significant reduction in cell numbers was observed (* *p* < 0.05). In (**D**), the same cell proliferation assay was carried out using the H358-shCtr and H358-shNQO1 cell lines. Here, again, a statistically significant reduction in cell proliferation was noted in the H358-shNQO1 cell line when compared to the H358-shCtr cell line (* *p* < 0.05). In (**B**,**D**) ns refers to no significant difference.

**Table 1 genes-14-00607-t001:** Cancer stem cell frequency in A549 and H358 cells with or without shNQO1 expression. Quantification of the number of cancer stem cells present in the examined cell populations was carried out utilizing an in vitro extreme limited dilution assay. The assay required plating limited dilutions of cells in low attachment conditions and examining wells for spheroid formation. Wells with at least one spheroid were counted as a positive well for the corresponding dilution. The results were then analyzed using Extreme Limited Dilution Analysis (ELDA) software. The resulting CSC frequencies demonstrate a marked increase in the CSC population found in A549-shCtr and H358-shCtr in comparison to their respective shNQO1 cell lines.

	40 Cells	120 Cells	360 Cells	720 Cells	CSC Frequency
A549- shCtr	2	0	19	18	1/455
A549- shNQO1	0	0	2	3	1/5659
H358- shCtr	0	0	1	1	1/14609
H358- shNQO1	0	0	1	0	1/29580

## Data Availability

The works published in this manuscript is freely available upon reasonable request, by the corresponding author.

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
