# Peer review of "NAD(P)H Quinone Oxidoreductase-1 Expression Promotes Self-Renewal and Therapeutic Resistance in Non-Small Cell Lung Cancer"

_genes, 2023, doi:10.3390/genes14030607_

Round 1
Reviewer 1 Report
Overall Summary of the Manuscript
This scientific article investigates the role of NAD(P)H: quinone oxidoreductase 1 (NQO1) in the maintenance of cancer stem-like cells in non-small cell lung cancer (NSCLC). The study found that NQO1 is essential for maintaining the CSC population and that its depletion increases the susceptibility of NSCLC cells to chemotherapeutic intervention. The results of the study suggest that combining NQO1-inhibitors with conventional chemotherapeutics may enhance anti-tumor effects and improve patient outcomes.
Overall Critique of the Manuscript
This manuscript provides an interesting exploration of how NAD(P)H quinone oxidoreductase-1 (NQO1) loss can reduce the self-renewing capabilities of non-small cell lung cancer (NSCLC) cells and enhance their susceptibility to chemotherapeutics. The authors have clearly presented the objectives of their research and have outlined the methods they used in their study. The results are backed up with evidence and are logically connected to the discussion and conclusion. The discussion and conclusion offer a clear understanding of the implications of their findings and provide potential future directions for research. The paper is well-written and the figures and tables are helpful in understanding the data. Overall, this is a well-written and informative manuscript that provides an interesting insight into the role of NQO1 in cancer drug resistance.
The methods section appears to be clear and concise, however there are a few areas that could be further clarified. First, there is not enough detail in the description of the western blots, such as the exact concentrations used. Second, the NQO1 activity assay section lacks detail regarding the specific steps taken during the assay process. Third, the extreme limited dilution assay could be more detailed, such as specifying the details of growth conditions.
Author Response
Dear Reviewer 1,
Thank you for taking the time to review our manuscript. The following changes were made to address your concerns:
- Detail on western blotting antibody concentrations/dilutions: We have updated the wording in the methods section for western blotting to clearly highlight the antibody dilutions for primary and secondary antibodies used in our western blot protocols
- NQO1 activity detail: The procedure for the NQO1 activity is detailed in the proprietary kit purchased from Abcam. Normally we would use methods that we previously published which indirectly measures NQO1 activity using a cytochrome c assay in the presence of NADPH or NADH. The kit uses similar principles and we found by measuring A549 and H596 cells as controls similar activity for these cell lines as in our previously published 2007 PNAS and 2016 Cancer Cell (bey et al) papers which are cited amongst the references in this manuscript.
- Growth conditions of the extreme dilutions: We have updated the incubator conditions for the extreme dilutions.
Reviewer 2 Report
NAD(P)H: quinone oxidoreductase 1 (NQO1) is generally overexpressed in non-small cell lung cancer (NSCLC) tumors. The authors have previously shown that depletion of NQO1 in NSCLC cell lines inhibited cell proliferation and anchorage-independent growth, decreased three-dimensional tumor spheroid, and reduced lung tumor xenograft growth. They have also noted that NQO1 depletion correlated with the loss of the cancer stem cell (CSC) marker aldehyde dehydrogenase (ALDH). In this study, they used a tumor spheroid model to test for the role of NQO1 in self-renewal and cisplatin-resistant phenotype in NSCLC cells.
They used stable knockdown of NQO1 in A549 and H358 cell lines with reduced NQO1 activity. They demonstrated that NQO1 depletion reduced primary, secondary, and tertiary tumor spheroid formation. Primary tumor spheroids from parental A549 cells expressed high levels of NQO1 and the putative CSC markers Sox2, Shh, and Nanog. They also demonstrated that loss of NQO1 expression reduced serial spheroid formation in A549 and H358 cell lines and re-expression of NQO1 in A549 cells rescues tumor-spheroid forming ability. They used the NQO1 null H596 cells to demonstrate that overexpression of NQO1 enhanced tumor spheroid formation. Using the limited dilution assays as an indicator of tumor-initiating cell frequency, they showed that A549 cells expressing NQO1 had a 12-fold increase in tumor-initiating cells versus with A549 cells lacking NQO1 expression. Finally, they showed that NQO1 knockdown in A549 and H358 cells rendered them more sensitive to cisplatin.
In general, the studies are well-written and designed and the right controls are included. The below comments will strengthen the manuscript:
It is will be useful to describe the different selected NCSLC lines in terms of mutations (p53, ras, EGFR…) and how they relate to lung tumor progression.
The Figure 1 legend does not describe Figure 1.
The authors should use in general more recent publications, for instance they list cancer statistics by Siegel RL et al. 2016 and 2019 as more recent ones up to 2023 have been published.
Are the A549 and H358 NQO1 knockdown cells just more sensitive to cisplatin? or also to other chemotherapeutic drugs such as paclitaxel?
Minor comment: line 23- role should be added after novel.
Author Response
Dear reviewer 2.
Thank you for taking the time to review our manuscript.
The following is our response to your queries:
- Discuss p53, Ras, EGFR status of cell lines and effect on tumor progression: We have added this in our discussion please see highlighted revisions in discussion section.
- Figure 1 legend error: We have corrected the legend for Figure 1.
- Update References: We have updated the Cancer Statistics reference to more clearly state the lung cancer cases expected in 2023.
- Are the knockdowns of NQO1 in A549 and H358 cells sensitive to other agents such as taxol: Although we tried other drugs the increase in sensitivity to cisplatin in NQO1-knockdowns was most significant. It is of note that the cells (A549 and H358) were also less sensitive to NQO1 mediated therapeutics such b-lapachone due to loss of NQO1, however this phenomenon is well known (and published by our lab and our collaborators labs) and was not shown in this manuscript.
- Add role after novel in line 23: We have made this correction.